# Runs of Homozygosity Revealed Reproductive Traits of Hu Sheep

**DOI:** 10.3390/genes13101848

**Published:** 2022-10-13

**Authors:** Yuzhe Li, Zitao Chen, Yifei Fang, Caiyun Cao, Zhe Zhang, Yuchun Pan, Qishan Wang

**Affiliations:** 1Hainan Institute, Zhejiang University, Yongyou Industry Park, Yazhou Bay Sci-Tech City, Sanya 572000, China; 2Department of Animal Science, College of Animal Science, Zhejiang University, 866# Yuhangtang Road, Hangzhou 310058, China; 3Hainan Yazhou Bay Seed Lab, Yongyou Industrial Park, Yazhou Bay Sci-Tech City, Sanya 572025, China; 4Department of Animal Science, School of Agriculture and Biology, Shanghai Jiao Tong University, 800# Dongchuan Road, Shanghai 200240, China

**Keywords:** runs of heterozygosity, Hu sheep, high-prolificacy breed, inbreeding coefficient

## Abstract

Hu sheep, a famous breed in the Taihu Basin, has the advantages of non-seasonal estrus, multiple fetuses, coarse feeding tolerance, and suitability for house feeding. Runs of homozygosity (ROHs) were found to be an effective tool to detect the animal population structure and economic traits. The detection of ROHs is beneficial for reducing the incidence of inbreeding as well as identifying harmful variants in the genome. However, there is a lack of systemic reports on ruminants in previous studies of ROHs. Here, we sequenced 108 Hu sheep, detected ROHs in Hu sheep to calculate their inbreeding coefficient, and selected genes of Hu sheep breeds within the ROH islands which are relevant to agricultural economic characteristics. Then, we compared the characteristics of the occurrences of SNPs between Hu sheep and other sheep breeds, and also investigated the distribution of the frequencies of SNPs within specific gene regions of Hu sheep breeds to select their breed-specific genes. Furthermore, we performed a comparative genome and transcriptome analysis in human and sheep breeds to identify important reproduction-related genes. In this way, we found some significant SNPs, and mapped these with a set of interesting candidate genes which are related to the productive value of livestock (*FGF9*, *BMPR1B*, *EFNB3*, *MICU2*, *GFRA3*), healthy characteristics (*LGSN*, *EPHA5*, *ALOX15B*), and breed specificity (*FGF9*, *SAP18*, *MICU2*). These results in our study describe various production traits of Hu sheep from a genetic perspective, and provide insights into the genetic management and complementary understanding of Hu sheep.

## 1. Introduction

Sheep (*Ovis aries*), as an important livestock species, have been domesticated for thousands of years. Since the Neolithic, artificial selection has led to sheep population diversity [1]. The Hu sheep is a famous sheep breed from the Taihu Plain in China, which has the advantages of high prolificacy, year-round estrus, good lactation performance, and fast growth [2]. Its reproduction ability is an economically valuable and high-profile trait. It is one of most prolific sheep breeds in the world. Generally, ewes produce lambs twice a year and litter sizes are usually 2–3, occasionally 7–8 [3]. Thus, knowledge of the genes involved in the ovulation rate and litter size will provide meaningful information for sheep breeding. In addition, the Hu sheep’s function of lactation is excellent. There are some researches on the nutritional value of Hu sheep milk. Chen et al. [4] reported that Hu sheep milk has unique lactic acid bacteria which has the advantage of direct vertical transmission and deserves further study. However, in recent years, most reports of microbes in milk are related to human milk or bovine milk, with few reports about other non-traditional sources of milk. Single-nucleotide polymorphisms (SNPs) are considered to be a kind of genetic marker existing widely in human and animal genomes, and regions with high homozygosity can be found effectively with intensive SNP detection [5]. ROHs, as continuous homozygous segments, are common in human and animal populations. ROH segments can be hereditary in a population and provide information about the demographic evolution of that population. Therefore, ROHs are used as a tool to characterize the degree of inbreeding depression within a population and to identify the candidate genes related to economically valuable traits. In summary, ROHs have various advantages in the detection of functional productive genes and guidance on livestock production. In recent research, ROHs have also been found in studies of livestock species. However, most of the studies on ruminants were concerned with cattle, and the research on ROHs needs to become more systematic and focused on more livestock breeds [6].

Therefore, the present study aimed to detect ROH patterns in Hu sheep populations, observe the degree of inbreeding in Hu sheep, and select candidate genes from within ROH islands which are related to breed-specific traits of Hu sheep. We investigated ROH islands which are related to characterized traits of Hu sheep using the sequencing data obtained by the Tn5-based method, which is a highly accurate, low-cost, and time-efficient low-coverage sequencing method [7]. We then explored the function of crucial Hu sheep genes by comparing them with other breeds. At the same time, we also used the results of human GWAS and TWAS to enrich the candidate genes, to further explore the expression of these genes in specific tissues and their association with disease development. Furthermore, we compared inbreeding coefficients which were calculated by different methods. Our study will provide fundamental evidence for Hu sheep breeding and complement present research on ROH detection in Hu sheep.

## 2. Materials and Methods

### 2.1. Data

In this research, a total of 108 Hu sheep (including 27 rams and 81 ewes) were analyzed. Their DNA samples were extracted from the sheep’s peripheral blood at Shanghai Hu sheep Conservation Farm, China. The protocol that we used for each sheep sequencing library was based on Tn5. [7] Tn5 library generation was implemented by TruePrep Tagment Enzyme Mix (TTE Mix) which contains transposase and two equimolar adaptors. The premix solution was mixed with DNA and incubated at 55 °C for 10 min to achieve DNA fragmentation and end splicing by mixing 10 μL of 5×TTBL (TruePrep Tagment Buffer L), 50 ng of DNA, 5 μL of TTE Mix V50 (TruePrep Tagment Enzyme) and ddH_2_O. The total volume of the reaction was 50 μL. 24 μL of fragment product was amplified by mixing 10 μL of 5×TAB (TruePrep Amplify Buffer), 5 Μl of PPM (PCR Primer Mix), 1 μL of TAE (TruePrep Amplify Enzyme), 5 μL of N5XX and 5 μL of N7XX (two index primers). The PCR program was as follows: 9 min at 72 °C, 9 cycles of 30 s at 98 °C, and then 30 s at 63 °C. Size selection and purification were performed using VAHTS^TM^ DNA Clean Beads. The purified and fragmented products obtained by the above steps is a library that can be sequenced. The libraries were sequenced on the Illumina 3000 platform.Whole-genome re-sequencing (WGS) data of sheep breeds (n = 248) were downloaded from the NCBI BioProject database under accession PRJNA624020 (mean 25.15×/sample). These sheep data included 16 Asiatic mouflon, 172 landraces, and 60 improved sheep from Asia, the Middle East, Africa, and Europe.

Raw data from re-sequencing and reduced sequencing were filtered using fastp software with default parameters. After filtering, the remaining reads were aligned to the sheep genome reference using the BWA tool [8]. Oar_rambouillet_v1.0 [9] was used as the reference genome for SNP calling in reduced sequencing data and re-sequencing data, which can detect higher SNP numbers than the method based on Oar v.4.0 [1]. SAM files were sorted by SAMtools software [10] with default parameters. Then, GATK4 software [11] was used for SNP calling of each individual. Population imputation was implemented by using BEAGLE 5.2. We then used bcftools-1.8 software [12] to filter the dosage R-squared for values lower than 0.9, with the command “bcftools filter -i ‘DR2 > 0.9’”. After that, PLINK v1.9 software [13] was adopted for quality control, with the command “plink --geno 0.1 --maf 0.05”. The remaining SNPs were used for further ROH analysis.

### 2.2. Definition of ROHs

PLINK v1.9 [13] was used for ROH detection, which uses a sliding window with a default number of SNPs to find the ROH regions within the sheep’s genome. To eliminate LD effects which were created by short homozygous segments, we set each ROH length threshold to 1 Mb. Specific parameter settings are based on the following: (1) each sliding window should contain 50 SNPs; (2) each ROH should have a sequence of more than a hundred consecutive SNPs; (3) the minimum density of each ROH segment was set to one SNP per 50 kb; (4) each ROH was allowed to contain up to five SNPs with missing genotypes and one SNP with heterozygous genotype due to genotyping error [14].

To reduce the occurrence of ROHs by chance, the minimum number of SNPs within a ROH was calculated using the formula below [15]:$$l=\frac{ln\frac{\alpha }{n_{s}n_{i}}}{ln(1-\bar{het})},$$
where α is the false-positive rate of ROH (set to 0.05), $n_{s}$ is the number of SNPs within the autosomes of an individual, $n_{i}$ is the total number of individuals within population i, and $\bar{het}$ is the mean heterozygosity of the total SNPs within population i. The minimum ROH length was set as 1 Mb. ROH segments were categorized based on their physical length into 1–5 Mb, 5–10 Mb, and ≥10 Mb, and identified as ROH_1–5Mb_, ROH_5–10Mb_, and ROH_>10Mb_ [5].

To explore the characteristics of Hu sheep, ROH detection was carried out both on the reduced sequencing data of Hu Sheep and the deep re-sequencing data of other breeds. Six kinds of sheep breeds (n = 75) were selected from the deep re-sequencing data by looking at the iSheep website (https://ngdc.cncb.ac.cn/isheep/, accessed on 16 May 2022), including Sishui Fur sheep, Tan sheep, Altay sheep, Bashibai sheep, Chinese Merino, and Duolang sheep. These sheep samples were all from China and have little geographical difference to Hu sheep, and their litter size (≤1 lamb per year) was significantly lower than Hu sheep.

### 2.3. Calculation of Inbreeding Coefficients

The genomic inbreeding coefficients for each individual were based on the ROHs ($F_{\mathrm{ROH}}$) in our study, and the equation that we used was as follows:$$F_{\mathrm{ROH}}=\frac{L_{\mathrm{ROH}}}{L_{\mathrm{aut}}},$$
where *L*_ROH_ is the total length of all ROHs in the genome of an individual and *L*_aut_ is the total 108 lengths of the autosomal genome covered by the SNPs (2.36 GB). $F_{\mathrm{SNP}1}$, $F_{\mathrm{SNP}2}$, and $F_{\mathrm{SNP}3}$ were calculated using “-ibd” in the GCTA software. These three parameters were estimated based on the information of SNPs [16,17].

### 2.4. Candidate Gene Annotation within ROH Islands

We used the frequency of each individual SNP in a ROH region to identify ROH islands. The SNPs with frequencies in the top 0.5% were identified as the candidate SNPs. The adjoining candidate SNPs form a region which is called a ROH island, and genes were further identified in these ROH islands [14]. We used an R package GALLO [18] to identify and annotate the genes in the ROH islands.

To identify the functions of candidate genes, KOBAS 3.0 (available online: http://bioinfo.org/kobas, accessed on 13 April 2022), a website for the enrichment of functional gene-related pathways and diseases, was used to perform the Kyoto Encyclopedia of Genes and Genomes (KEGG) and Gene Ontology (GO) enrichment. The annotation of genes within the ROH islands was based on the Animal Quantitative Trait Loci (QTL) Database [19].

In addition, to further explore the potential roles of the candidate genes within ROH islands, we collected genome-wide association study (GWAS) summary statistics and transcriptome-wide association study (TWAS) research for related traits in humans to annotate candidate genes with potential relevance to human disease. The results of GWAS and TWAS were collected from the webTWAS database [20].

## 3. Results

### 3.1. DNA Sequencing and Genetic Diversity

Through the Tn5-based low-coverage sequencing method, a total of 1.36 billion clean reads were found, and the mean was 12.63 million clean reads (with 0.29% mean sequencing coverage) per sheep (Figure 1 and Appendix A). After imputation and filtering the invalid data, we obtained 14.18 million whole-genome single-nucleotide polymorphisms in 108 Hu sheep at a mean depth of 22.39×. A total of 6.55 million (55.54%) SNPs were found in gene regions, and there were 48,657 SNPs in exon regions and 7668 SNPs in untranslated regions. The exons made up only 0.7% of the SNPs in gene regions, and this could be due to the uncomplete annotation information and high-depth sequencing after imputation.

### 3.2. Statistics of Inbreeding Coefficients

The inbreeding coefficients based on different methods are shown in Table 1. Among the three average inbreeding coefficient estimates based on different lengths of ROH segments, *F*_ROH1–5Mb_ is significantly larger than *F*_ROH>10Mb_ and *F*_ROH5–10Mb_.

The inbreeding coefficients obtained from the different physical lengths of the ROH fragments vary greatly; *F*_ROH>10Mb_ and *F*_ROH5do10Mb_ were much smaller than *F*_ROH1–5Mb_ (Table 2). The correlation among ROH-based methods was strong, and the highest correlation (0.965) was found between *F*_ROH-all_ and *F*_ROH1–5Mb_. ROH segments of 1–5 Mb in length made up 95.49% of all ROH segments. The weakest correlation (0.649) among the ROH-based estimates was between *F*_ROH1–5Mb_ and *F*_ROH5–10Mb_, which was 0.649.

The correlation of inbreeding coefficients obtained from the SNPs information has some differentiation; $F_{\mathrm{SNP}2}$ was weakly correlated with $F_{\mathrm{SNP}1}$ and $F_{\mathrm{SNP}3}$_._ Except for $F_{\mathrm{SNP}2}$, other $F_{\mathrm{SNP}}$ had weak correlations with ROH-based inbreeding coefficients.

### 3.3. Distribution of ROHs

Through ROH detection, 5904 ROH segments were found in 108 Hu sheep (Figure 2). The longest segment (24.11 Mb) and the shortest segment (1 Mb) were both found in chromosome 3, which contains 114,373 SNPs and 4466 SNPs, respectively. The statistics of ROH numbers and length classified by the physical length of the ROH are shown in Table 3.

In terms of physical length, the short ROH (1–5 Mb) segments made up the majority (95.49%) of the whole ROH length, and ROH segments longer than 10Mb accounted for just 0.73% of the whole ROH length.

### 3.4. Gene Annotation

Through the method mentioned above, nineteen ROH islands were found with 59,268 SNPs and 142 genes. The position of each ROH island and the number of SNPs within each ROH island are shown in Table 2. The longest ROH island is in chromosome 10, which was found between 37,178,926 and 41,620,003 bp. This island contained 17 genes and may be the most relevant region for functional expression in Hu sheep.

We then annotated all genes within potential ROH islands, and found that the majority of them were correlated with some economically important traits, particularly a lot of reproductive traits (Table 4 and Figure 3A). The genes that we detected had connections with Hu sheep reproduction (*FGF9*, *BMPR1B*, *EFNB3*), milk (*LGSN*, *OCA2*, *HERC2*), meat (*MICU2*, *HAO1*, *OCA2, SPATA5*), and wool traits (*GFRA3*, *CDC23*, *CDC25C*). Then, based on the sheep QTL database, we found four QTLs within these ROH islands which were related to reproductive traits.

Through the function enrichment of candidate genes, we found that nineteen KEGG pathways and seven GO terms were significantly correlated with human disease processes or biosynthesis pathways, specifically related to breast disease (Appendix A).

To further explore the role of reproduction-related genes in animal function and their relevance to disease development, we collected the results of GWAS summary statistics, and TWAS research for related traits in humans from the webTWAS database. Thirteen candidate genes associated with reproduction are shown in Appendix A. These genes were mainly associated with human reproductive organs (testis, vagina) and vascular/heart problems. This result indicates that reproduction-related genes of Hu sheep express similar effects in humans and play an important role in human health.

Furthermore, to find breed-specific genes in Hu sheep, we selected sheep breeds with weak reproductive ability in the re-sequencing data and calculated the frequencies of their SNPs. By comparing characteristics of the distribution of SNPs in Hu sheep with other sheep, we found that these sheep shared 8804 SNPs and 19 genes with Hu sheep. After eliminating the common SNPs, we enriched the Hu sheep’s unique genes and observed the distribution of the density of SNPs within the three reproduction-related gene regions with the highest frequency of ROHs (Figure 3B). These genes (*FGF9*, *SAP18*, *MICU2*) may be correlated with specific high-profile reproductive traits of Hu sheep.

## 4. Discussion

### 4.1. Pattern of ROHs

Through ROH detection, the lengths of the ROHs were mostly found to be within the region of 1–5 Mb. The number of large segments is much lower, especially the segments longer than 10 Mb. This indicates that the level of inbreeding among these Hu sheep is low and they are hardly influenced by recent inbreeding events. However, it is worth mentioning that ROH length is not only dependent on inbreeding events. There are reports claiming that the formation and evolution of ROHs is a random event due to the dynamic recombination and randomness in the process of gamete formation. Furthermore, decreased population size and bottleneck events can also affect the characteristics of short segments of less than 4 Mb [21].

The pattern of the ROHs in our study has specificity in different chromosomes, and the number of ROHs was affected by the chromosome’s physical length. Nandolo et al. [22] also reported a similar result, and they found that ROHs tend to enrich on the specific chromosomes which have high levels of support for IBD accumulation.

In addition, the coverage of ROHs is also relevant to the functional genes of animals. Chr 6 and Chr 10 displayed the highest coverage of ROHs within the Hu sheep population, and annotated the largest number of candidate genes (Figure 2). This result could be due to artificial selection, which is predisposed to improve the level of accumulation of ROHs in certain gene regions and allows the expression of relevant economic traits.

### 4.2. Inbreeding Level within the Population

The inbreeding coefficients were calculated based on ROH segments. $F_{\mathrm{ROH}}$ can be more efficient than a traditional pedigree that is manually recorded because human-recorded information is prone to error. Furthermore, inbreeding coefficients based on ROHs and SNPs can reflect more information about the realized homozygous site than traditional pedigrees [23].

The values of other genomic inbreeding coefficients ($F_{\mathrm{SNP}1}$, $F_{\mathrm{SNP}2}$, and $F_{\mathrm{SNP}3}$) which were calculated using GCTA software were not close to reality, and some of the coefficients were even higher than 1. This is because of the way that the GCTA software calculates inbreeding coefficients based on the genomic relatedness matrix, and the result may be affected by the population size and depth of sequencing. $F_{\mathrm{GCTA}}$ was not as accurate as $F_{\mathrm{ROH}}$ as a measure of genomic breeding [24]. Furthermore, reduced sequencing was used in this study, which only covered 0.3% of the whole genome. Through the sequencing and imputation method, some regions were incorrectly identified as homozygous segments, and produced false-positive results.

### 4.3. Functional Enrichment Analyses

In the ROH island which we identified, several candidate genes were annotated and were mainly related to reproduction, milk production, meat, and some economic traits. We annotated a series of genes which were associated with the Hu sheep’s high reproduction traits (Figure 3A). According to the sheep QTL database, we found that some genes map to QTLs related to testes and lambs (ID = 12923, ID = 154661, ID = 154663, ID = 130456). These genes have been previously reported in other animals and are closely related to the expression of reproductive traits in many animals. Previous studies have shown that embryo development is an important factor influencing reproduction. Zygotic genome activation initiates the expression of the parental genome, during which any misbehavior may terminate embryonic development. Mutations in *GJB6* may lead to ectodermal dysplasia [25]. *MPHOSPH8* can make *Fam208a* play a dual role during zygotic cleavage and early embryonic development through interaction with it [26]. It was reported that *FGF9* can affect reproduction in several respects, including the ovarian function [26], the development of sperm [27], maintaining pregnancy [28], and male sex development [29]. These results may prove that *FGF9*, *GJB6*, and *MPHOSPH8* play a similar role in sheep and determine sheep reproductive ability by affecting ovarian and embryo development as well as sperm formation.

We also identified some genes which were reported as reproductive-relative genes in sheep or other livestock. *BMPR1B* has been reported in many sheep breeds, and it was a major gene affecting litter size [30]. *EFNB3* was regulated by two up-regulated lncRNAs and one down-regulated lncRNAs, and these lncRNAs were involved in Hu sheep reproduction [31]. Feng et al. [32] found *SAP18* had significantly different expression in the early stage of chickens’ gonad development. *ATP12A* was related to the development of trophectoderm in bovines [33].

By comparing with other sheep with poor reproductive ability, we found that Hu sheep and other sheep breeds shared *EFNB3*, *GJB6*, and *SAP18*. This phenomenon could confirm that these three genes are double-regulated in the expression of reproductive traits in animals [25,31,33]. In the sections with significant density differences, *FGF9*, *SAP18*, and *MICU2* were screened, which may explain the unique trait of high prolificacy in Hu sheep.

We then performed a functional enrichment analysis for candidate genes with human GWAS and TWAS results. We found that *LGSN* can also have a high expression in human tissues and may be associated with some human autoimmune diseases. We also noted that *FGF9, BMPR1B*, *EFNB3*, *GJB6*, and *SAP18,* which were identified as the reproduction-related genes of Hu sheep, are highly expressed in human reproductive organs, such as testes, ovaries, and so on. These results explain the functions of these candidate genes and suggest that they may express similar functions in humans.

Sheep milk is another high-profile trait of Hu sheep. In KEGG pathway enrichment analyses, there are five genes in breast disease pathways. *LGSN*, *OCA2*, and *HERC2* reportedly mediated resistance to breast cancers [34,35,36]. In addition, we found some candidate genes which are associated with livestock meat traits (*MICU2*, *HAO1*, *OCA2*, *SPATA5*) and some genes related to wool traits (*GFRA3*, *CDC23*, *CDC25C*). These results can explain the good meat quality, milk, and wool-production performance of Hu sheep.

## 5. Conclusions

In the present study, we sequenced Hu sheep and detected ROHs in Hu sheep to calculate their inbreeding coefficient, and selected ROH islands that contain the candidate genes related to breed-specific traits of Hu sheep breeds. We found that the population of Hu sheep was not significantly affected by historical inbreeding events. Then, we identified the functional genes within the ROH islands and conducted enrichment analyses. The results showed that the vast majority of genes within the ROH island were related to human disease and biologically related pathways. We also performed a comparative genome analysis in different sheep breeds and identified that *FGF9*, *SAP18*, and *MICU2* may be the Hu sheep’s breed-specific reproduction-related genes. Furthermore, we used human GWAS and TWAS results to annotate candidate genes and observe the expression of these genes in organs, tissues, and disease development to identify important reproduction-related genes. We found that *FGF9*, *SAP18*, *MICU2*, *BMPR1B*, *EFNB3*, *GJB6*, and *LATS2* are highly expressed in human reproductive organs and may play an important role in the reproductive function of sheep and humans. These findings may provide new insights into the characteristics of Hu sheep and bring new strategies for future genetic improvements in sheep.

## Figures and Tables

**Figure 1 genes-13-01848-f001:**
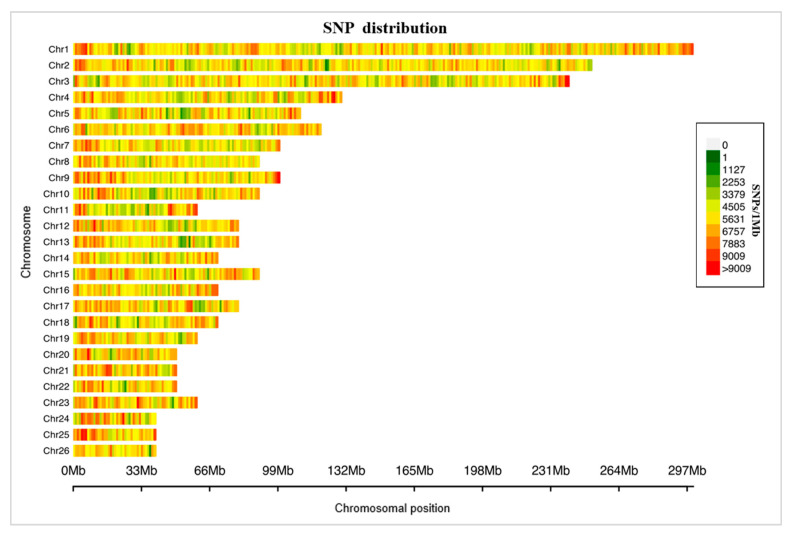
Distribution of the sequenced SNPs on all chromosomes. The *y*-axis represents chromosomes, and the *x*-axis represents the corresponding chromosomal position (Mb). Different colors of each 1-Mb genome block denote the number of SNPs.

**Figure 2 genes-13-01848-f002:**
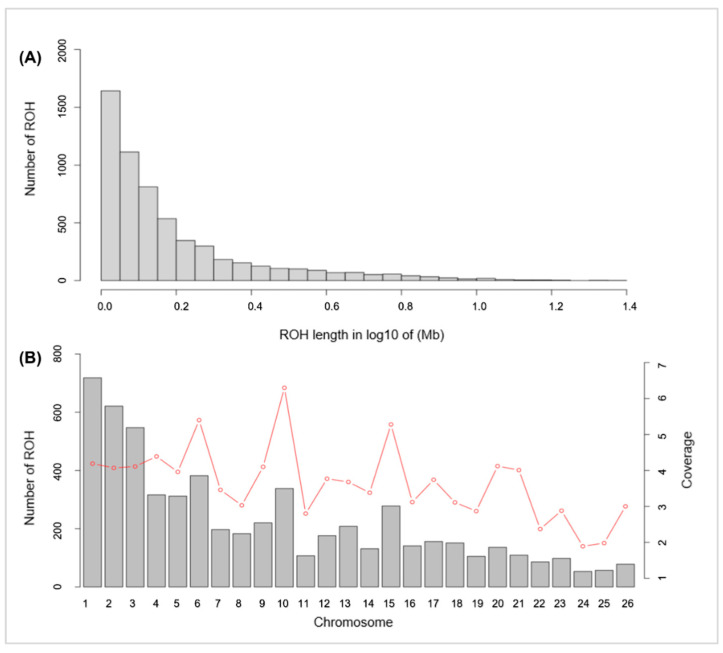
Distribution of the ROHs. (**A**) Length distribution of the ROHs. *X*-axis represents the length of the ROH (Mb) using a base-10 log scale. *Y*-axis represents the number of ROH for different ROH lengths. (**B**) Bars and the red dotted line represent the number of ROH and the ROH coverage, respectively, on each chromosome.

**Figure 3 genes-13-01848-f003:**
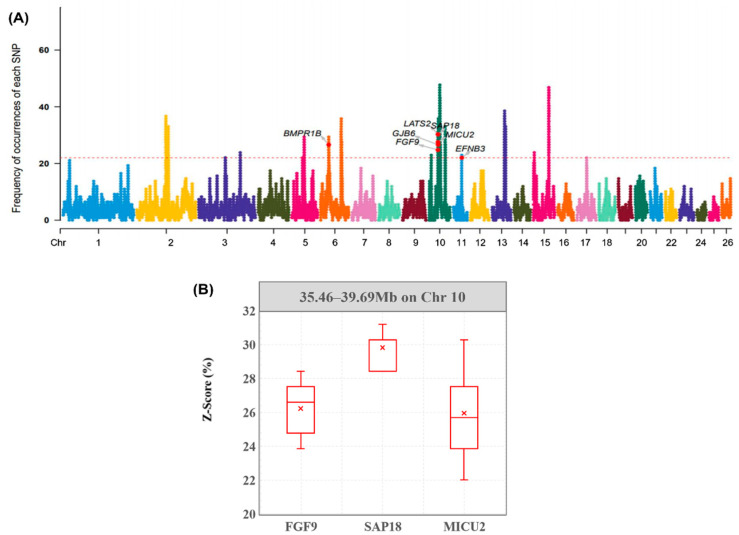
Frequency of occurrences of each SNP within ROH regions among all individuals. (**A**) Horizontal red line represents the 20% threshold. Genes related to reproduction quality hare identified. (**B**) The plot represents the frequency distribution of SNPs within unique gene regions of Hu sheep. The *X*-axis represents three different candidate genes within the ROH island. The *Y*-axis represents the frequency of occurrences of each SNP within the gene annotation interval.

**Table 1 genes-13-01848-t001:** Descriptive statistics for the seven types of inbreeding coefficient estimates calculated from the identified ROHs (*F*_ROH-all_, *F*_ROH1–5Mb_, *F*_ROH5–10Mb_, and *F*_ROH>10Mb_), and the SNP information (*F*_SNP1_, *F*_SNP2_, and *F*_SNP3_).

Inbreeding Coefficient	Mean	Min	Max	SD
** *F* _ROH1–5Mb_ **	0.034	0.001	0.113	0.028
** *F* _ROH5–10Mb_ **	0.014	0.002	0.065	0.014
** *F* _ROH>10Mb_ **	0.010	0.004	0.026	0.006
** *F* _ROH-all_ **	0.041	0.001	0.176	0.040
** *F* _SNP1_ **	0.921	0.685	1.109	0.086
** *F* _SNP2_ **	0.921	0.882	0.964	0.019
** *F* _SNP3_ **	0.921	0.817	1.002	0.038

**Table 2 genes-13-01848-t002:** Correlation coefficients (lower panel) among seven types of inbreeding coefficient estimates (*F*_ROH-all_, *F*_ROH1–5Mb_, *F*_ROH5–10Mb_, and *F*_ROH>10Mb_, *F*_SNP1_, *F*_SNP2_, and *F*_SNP3_).

Correlation	*F* _ROH1–5Mb_	*F* _ROH5–10Mb_	*F* _ROH>10Mb_	*F* _ROH-all_	*F* _SNP1_	*F* _SNP2_	*F* _SNP3_
** *F* _ROH1–5Mb_ **	1						
** *F* _ROH5–10Mb_ **	0.649	1					
** *F* _ROH>10Mb_ **	0.654	0.674	1				
** *F* _ROH-all_ **	0.965	0.817	0.768	1			
** *F* _SNP1_ **	−0.423	−0.067	−0.327	−0.356	1		
** *F* _SNP2_ **	0.675	0.277	0.474	0.610	−0.652	1	
** *F* _SNP3_ **	−0.318	−0.009	−0.257	−0.257	0.982	−0.499	1

**Table 3 genes-13-01848-t003:** Descriptive statistics for ROHs of different lengths (Mb): 1–5, 5–10, >10, and >1.

ROH Length (Mb)	ROH Number	Percent (%)	Mean Length (Mb)	Standard Deviation
1–5	5638	95.49	1.55	0.74
5–10	223	3.78	6.62	1.21
>10	43	0.73	12.86	3.25
Total (>1)	5904	100	1.82	1.58

**Table 4 genes-13-01848-t004:** List of ROH islands observed in the Hu sheep population.

Chr	Start (MB)	End (MB)	No. SNPs	No. Genes	Candidate Genes	Gene Functions
1	25.96	26.05	91	1	-	-
2	116.88	119.76	6054	8	*OCA2*, *LGSN*	Meat and milk
2	127.24	128.51	4739	0	-	-
3	107.12	107.77	851	5	*MERTK*, *ZC3H8*	Meat and heath
3	167.83	168.54	791	0	-	-
5	43.42	43.83	302	1	-	-
5	49.58	50.67	1518	20	*GFRA3*	Health
6	32.26	33.36	4024	4	*BMPR1B*	Reproduction
6	82.18	85.35	3911	1	*EPHA5*	Health
10	35.46	39.69	12137	17	*FGF9*, *MICU2*	Reproduction
10	42.23	45.97	11643	0	-	-
10	64.55	65.82	2732	0	-	-
11	34.06	34.77	517	36	*ALOX15B*	Reproduction
13	49.23	50.55	2321	2	*HAO1*	Meat
13	53.31	54.35	1383	30	*GINS1*	Health
15	3.23	3.45	968	0	-	-
15	61.59	62.62	562	0	-	-
17	38.73	38.89	332	17	*SPATA5*	Meat

## Data Availability

The data presented in this study are available on request from the corresponding author. The data are not publicly available due to privacy restrictions.

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
