# Peer review of "Runs of Homozygosity Revealed Reproductive Traits of Hu Sheep"

_genes, 2022, doi:10.3390/genes13101848_

Round 1

Reviewer 1 Report

The proposed manuscript "Runs of Homozygosity Revealed Agriculture Characteristics of Hu Sheep" is correctly written.
The Introduction chapter is sufficiently informative.
In the Material and Methods chapter, the research itself is well described.
The Discussion chapter is not too long and could be more detailed, but I would not insist on it.
References 24 and 34 in the bibliography should be corrected.

Author Response

Cover letter

October 6, 2022

The Editor

Genes Journal

Dear Ms. Zinnia Liu,

Thank you very much for giving us the opportunity to revise this manuscript. According to yours and the reviewers’ comments and suggestions, our manuscript has been revised carefully and completely. We have already corrected References in the bibliography as well as added some details in chapters. These constructive comments and revision work improved this manuscript obviously.

We hope you will find that our contribution is suitable for publication in Genes Journal, and we are looking forward to your comments.

Sincerely yours,

Qishan Wang, Ph.D.

Department of animal breeding and genetics

Zhejiang University

E-mail: wangqishan@zju.edu.cn

Reviewer 2 Report

The research talks about the relationship Runs of Homozygosity on some Characteristics of Hu Sheep. The research is interesting and shows results that can help in the understanding of these mechanisms. However, some adjustments need to be made before final publication. I do not consider myself an expert in the English language but I suggest reviewing some terms and words in the document. Therefore, I recommend that it be minor revisions.

Title: I recommend reviewing and trying to use a more appropriate title for this research. For example, how correct is it to use the term “Agriculture Characteristics”.

Abstract:

Line 12: I suggest changing “estrus in all seasons” to “non-seasonal”

Line 14: What do you mean by the term "long-time artificial and natural selection"?

Introduction:

General recommendations: I recommend expanding this section. Also describe the "SNPs".

Line 34: I recommend removing “etc”

Line 54: I recommend adjusting and expanding the objective “the present study aimed to detect Hu sheep ROH patterns in populations” according to the research approach.

Materials and methods

In general, it is necessary to add a section where the data processing and its statistical analysis, significance, etc. I mean, described in detail.

 Results:

I suggest restructuring this section.

Citations are added in this section “Then we calculated frequencies of SNPs in sequencing data (Fig. 1 and Table S1) and annotated these SNPs by ANNOVAR software [16].” This fraction would correspond better to statistical analysis.

Line 146-148: same case as the previous comment, in this section do not append citations or report your results “The 146 mean ?ROH in this Hu sheep population is 0.041, which was smaller than previous value 147 (0.093) calculated by Tao et al . [17] and it was similar with the value (0.048) that Zhang et al. 148 [18] have reported.”

 Discussion

I suggest discussing your results according to how you reported the most important findings in the results section.

Line 233: in this section it is not recommended to use figures, this would go in the results section (Fig 2).

Conclusions

I suggest trying to make it more specific and clear.

Author Response

Cover letter

October 6, 2022

The Editor

Genes Journal

Dear Ms. Zinnia Liu,

Thank you very much for giving us the opportunity to revise this manuscript. According to yours and the reviewers’ comments and suggestions, our manuscript has been revised carefully and completely. These constructive comments and revision work improved this manuscript obviously.

The comments raised by reviewers have been responded point-by-point. In this revision, we exchanged our title. We have replaced and deleted the incoherent words in the chapters of abstract as well as introduction. In the introduction section, we added an introduction to SNPS and a description of the purpose of the experiment. In the description of materials and methods, we added the specific content of data processing. In results and discussion section, we removed some unreasonable quotations and descriptive utterances. Meanwhile, we moved Fig2 from the discussion section to the results section and revised the content of the summary section. Furthermore, we also corrected References in the bibliography.

We hope you will find that our contribution is suitable for publication in Genes Journal, and we are looking forward to your comments.

Sincerely yours,

Qishan Wang, Ph.D.

Department of animal breeding and genetics

Zhejiang University

E-mail: wangqishan@zju.edu.cn

Editor and Reviewer comments:  

Reviewer #2: The research talks about the relationship Runs of Homozygosity on some Characteristics of Hu Sheep. The research is interesting and shows results that can help in the understanding of these mechanisms. However, some adjustments need to be made before final publication. I do not consider myself an expert in the English language but I suggest reviewing some terms and words in the document. Therefore, I recommend that it be minor revisions.

Author: Thank you for your careful work and thoughtful suggestions that have helped improve our paper substantially.

Title: I recommend reviewing and trying to use a more appropriate title for this research. For example, how correct is it to use the term “Agriculture Characteristics”.

Author: Thanks for your suggestion. We have revised the title. Please see P1/L1-2.

“Runs of Homozygosity Revealed Reproductive Traits of Hu Sheep”

Abstract:

Line 12: I suggest changing “estrus in all seasons” to “non-seasonal”

Line 14: What do you mean by the term "long-time artificial and natural selection"?

Author: Thanks for your reminder. Revised. Please see P1/L12-18.

Introduction:

General recommendations: I recommend expanding this section. Also describe the "SNPs".

Author: Thanks for your reminder. Added. Please see P2/L47-49.

“Single nucleotide polymorphisms (SNPs) are considered as a kind of genetic marker widely existing in human and animal genomes, and the regions with high homozygosity can be found effectively with intensive SNPs detection [5]. ROH, as continuous homozygous segments, are common in human and animal populations.”

Line 34: I recommend removing “etc”

Author: Thanks for your reminder. Revised. Please see P1/L37.

Line 54: I recommend adjusting and expanding the objective “the present study aimed to detect Hu sheep ROH patterns in populations” according to the research approach.

Author: Thanks for your reminder. Added. Please see P1/L59-61.

“Therefore, the present study aimed to detect Hu sheep ROH patterns in populations, observe the situation of inbreeding of Hu sheep and select the candidate genes within ROH island which are related to breed-specific traits of Hu sheep.”

Materials and methods

In general, it is necessary to add a section where the data processing and its statistical analysis, significance, etc. I mean, described in detail.

Author: Thanks for your reminder. Added. Please see P2/L84-95.

“Raw data from re-sequencing and reduced sequencing was filtered using fastp software with default parameters. After filtering, the remaining reads were aligned to sheep genome reference by BWA tool [8]. Oar_rambouillet_v1.0 [9] was used as the reference genome for SNP calling in reduced sequencing data and resequencing data, which can detect higher SNP number than the method based on Oar v.4.0 [1]. SAM files were sorted by SAMtools software [10] with default parameter. Then, GATK4 software [11] was used for SNP calling of each individual. Population imputation was implemented by using BEAGLE 5.2. Then we used bcftools-1.8 software[12] to filter the dosage R-squared which is lower than 0.9 with the command “bcftools filter -i 'DR2>0.9'”. After that, PLINK v1.9 software [13] was adopted for quality control with the command: “plink --geno 0.1 --maf 0.05” The remaining SNPs were used for further ROH analysis.”

 Results:

I suggest restructuring this section.

Citations are added in this section “Then we calculated frequencies of SNPs in sequencing data (Fig. 1 and Table S1) and annotated these SNPs by ANNOVAR software [16].” This fraction would correspond better to statistical analysis.

Author: Thanks for your reminder. Revised. Please see P4/L153-158.

Line 146-148: same case as the previous comment, in this section do not append citations or report your results “The 146 mean ?ROH in this Hu sheep population is 0.041, which was smaller than previous value 147 (0.093) calculated by Tao et al . [17] and it was similar with the value (0.048) that Zhang et al. 148 [18] have reported.”

Author: Thanks for your reminder. Revised. Please see P4-5/L168-173.

 Discussion

I suggest discussing your results according to how you reported the most important findings in the results section.

Author: Thanks for your reminder. Revised. Please see P10/L287-341.

Line 233: in this section it is not recommended to use figures, this would go in the results section (Fig 2).

Author: Thanks for your reminder. Revised. Please see P6/L199-201.

“In the ROH island which we identified, several candidate genes were annotated and mainly related to reproduction, milk production, meat and some economic traits. We annotated a series of genes which were associated with Hu sheep’s high reproduction traits (Fig. 3A). According to the Sheep QTL database, we found that some genes map to QTLs related to testes and lambs (ID = 12923, ID = 154661, ID = 154663, ID = 130456). These genes have been previously reported in other animals and are closely related to the expression of reproductive traits in many animals. Previous studies have showed that embryo development is an important factor influencing reproduction. Zygotic genome activation initiates the expression of the parental genome, during which any misbehavior may terminate embryonic development. Mutations in GJB6 may led to ectodermal dysplasia [25]. MPHOSPH8 can make Fam208a play a dual role during zygotic cleavage and early embryonic development by interaction with it [26]. FGF9 was reported that can affect reproduction in several aspects, including the ovarian function [26], the development of sperm maintaining pregnancy[28] and male sex development [28].. These results may prove that FGF9, GJB6 and MPHOSPH8 play a similar role in sheep and determine sheep reproductive ability by affecting development of ovarian and embryo as well as sperm formation.

We also identified some genes which were reported as reproductive relative genes in sheep or other livestock. BMPR1B has been reported in many sheep breeds, and it was a major gene affecting litter size [30]. EFNB3 was regulated by two up-regulated lncRNAs and one down-regulated lncRNAs, and these lncRNAs were involved in Hu Sheep reproduction [31]. Feng et al. [31] found SAP18 had significantly different expression at chicken’s the early stage in gonad development ATP12A was related to bovine’s development of trophectoderm [33].

By comparing with other sheep with poor reproductive ability, we found that Hu sheep and other sheep breeds shared EFNB3, GJB6 and SAP18. This phenomenon could confirm that these 3 genes have double regulation in expression of reproductive traits in animal [25], [31], [33]. In the sections with significant density differences, FGF9, SAP18 and MICU2 were screened, which may explain the unique trait of high prolificacy in Hu sheep

Then we did functional enrichment analysis for candidate gene with human GWAS and TWAS results. We found that LGSN can also have a high expression in human tissues and be associated with some human autoimmune disease. We also noted that FGF9, BMPR1B, EFNB3, GJB6 and SAP18, which were identified as the reproduction-related genes of Hu sheep, are highly expressed in human reproductive organs, such as testes, ovaries, and so on. These results explained the functions of these candidate genes and suggested that they may express similar functions in humans.

Sheep milk is another high-profile trait for Hu sheep. In KEGG pathway enrichment analyses, there are 5 genes in breast diseases pathways. LGSN, OCA2, and HERC2 were reported that can mediated resistance in breast cancers [34], [35], [36]. In addition, we also found some candidate genes which are associated with livestock’s meat traits (MICU2, HAO1, OCA2, SPATA5) and some wool traits relative genes (GFRA3, CDC23, CDC25C). These results can explain the good meat quality, milk and wool production performance of Hu sheep.”

Conclusions

I suggest trying to make it more specific and clear.

Author: Thanks for your reminder. Revised. Please see P11/L346-362.

“In the present study, we sequenced Hu sheep and detect ROH in Hu sheep to calculate their inbreeding coefficient and selected ROH islands that contain the candidate genes related to breed-specific traits of Hu sheep breeds. We found that the population of Hu sheep was not significantly affected by historical inbreeding events. Then, we identified the functional genes within ROH islands and conducted enrichment analyses to enrichment analysis. The results showed that the vast majority of genes within ROH island were related to human disease and biological related pathways. We also did comparative genome analysis in different sheep breeds and identified that FGF9, SAP18, MICU2 may be Hu sheep’s breed-specific reproduction-related genes. Furthermore, we used human GWAS and TWAS results to annotate candidate genes and observe the expression of these genes in organs, tissues and disease development to identify important reproduction-related genes. We found that FGF9, SAP18, MICU2, BMPR1B, EFNB3, GJB6 and LATS2 had high expression in human’s reproductive organs and may play an important role in reproductive function of sheep and human. These findings may provide new insights into the characteristics of Hu sheep and bring new strategies for future sheep genetic improvement.”
